# Proton Therapy in the Adolescent and Young Adult Population

**DOI:** 10.3390/cancers15174269

**Published:** 2023-08-25

**Authors:** Safia K. Ahmed, Sameer R. Keole

**Affiliations:** Department of Radiation Oncology, Mayo Clinic Arizona, Phoenix, AZ 85054, USA; keole.sameer@mayo.edu

**Keywords:** adolescents, young adults, proton therapy, cancer

## Abstract

**Simple Summary:**

We discuss the utility of proton beam radiation therapy for the treatment of common cancers seen in the adolescent and young adult population.

**Abstract:**

Background: Adolescent and young adult cancer patients are at high risk of developing radiation-associated side effects after treatment. Proton beam radiation therapy might reduce the risk of these side effects for this population without compromising treatment efficacy. Methods: We review the current literature describing the utility of proton beam radiation therapy in the treatment of central nervous system tumors, sarcomas, breast cancer and Hodgkin lymphoma for the adolescent and young adult cancer population. Results: Proton beam radiation therapy has utility for the treatment of certain cancers in the young adult population. Preliminary data suggest reduced radiation dose to normal tissues, which might reduce radiation-associated toxicities. Research is ongoing to further establish the role of proton therapy in this population. Conclusion: This report highlights the potential utility of proton beam radiation for certain adolescent young adult cancers, especially with reducing radiation doses to organs at risk and thereby potentially lowering risks of certain treatment-associated toxicities.

## 1. Introduction

Cancer is the primary cause of health-related death in the adolescent and young adult (AYA) population, defined as patients aged 15–39 years [1,2,3]. AYA patients are at high risk of developing long-term radiation-associated toxicities given their young age and anticipated decades of survivorship [4,5,6]. These toxicities include cognitive deficits, cardiovascular disease, hormone deficiencies, fertility difficulties, organ dysfunction, and secondary malignancies [4,5,6]. Furthermore, treatment outcomes and survivorship care for some AYA cancers have remained stagnant over time despite significant advances noted for pediatric and older adult patients [7,8,9]. Factors likely contributing to this include increased financial burdens, inadequate insurance coverage, decreased access to specialized cancer care, and under-representation in clinical trials seen with AYA patients [1,8].

As such, when AYA patients are treated with radiation therapy, it is essential to minimize organ exposure to radiation and the integral dose to maximize health-related quality of life [10]. Proton beam radiation therapy (PBT) can provide a strong dosimetric advantage by reducing entry dose and eliminating exit dose, compared to photon radiation therapy (XRT) [11]. Additionally, treatment efficacy is not sacrificed with PBT, thus potentially optimizing the therapeutic ratio for radiation treatments [11]. However, there are limited randomized data comparing treatment-associated toxicities with PBT to XRT, particularly in the AYA population. AYA patients are also less likely to receive insurance approval for PBT, further limiting research efforts for PBT in the population [12]. The purpose of this article is to discuss the utility of PBT for common cancers treated with radiation therapy in the AYA population.

## 2. Central Nervous System Tumors

CNS tumors account for 4.4% of all cancers in AYA, ranking it as the 11th most common cancer type in AYA patients aged 15–39 years [13]. AYA patients with brain tumors appear to have inferior outcomes, as compared to younger (<14 years old) patients, due to multiple reasons. AYA patients are less likely to have protocol-directed treatment and are more likely to be uninsured or underinsured [1,14].

The target volume should be the same when treating with either PBT or XRT. PBT offers potential advantages in dose to non-target tissues. In patients with CNS tumors, this non-target dose reduction can lead to less risk of cognitive impairment, hormonal dysfunction and secondary malignancies [15].

Several studies have compared the cognitive side effects of proton therapy to those of conventional radiation therapy in patients with brain tumors. Pediatric patients treated with PBT were found to have higher gull-scale IQ and processing speed relative to XRT [16]. In the PBRT group, no change in IQ over time was identified (*p* = 0.130), whereas in the XRT group, IQ declined by 1.1 points per year (*p* = 0.004) [17]. This difference was more pronounced in patients treated with focal RT fields [18].

Higher radiation doses to diencephalic structures, such as the hypothalamus and pituitary, can lead to increased risk of endocrinopathy in children and young adults with brain tumors [19]. PBT appears to reduce these risks. Eaton et al. published that PBT decreased risks of hypothyroidism (23% versus 69% with photon therapy), sex hormone deficiency (3% versus 19% with photon therapy), and requirement for any endocrine replacement therapy (55% versus 78% with photon therapy). Patients treated with PBT also had a greater height standard deviation score (i.e., taller) compared to patients treated with photon therapy. There was no significant difference in the incidence of precocious puberty, adrenal insufficiency or growth hormone deficiency between patients treated with PBT and photon therapy [20]. In a series of 118 patients with medulloblastoma, photon therapy was associated with a higher incidence of primary hypothyroidism compared to PBT: 28% versus 6%. This was felt to be due to lower dose exposure to the thyroid gland and the pituitary gland after spine and posterior fossa boost radiation [21].

The incidence of radiation-induced brain tumors (RIBTs) following initial cranial irradiation has been well-studied with XRT and modeled with PBT, although there are some limited clinically based estimates with the latter. With XRT, the relative risk of RIBT is between 6 and 10 with a latency period of 6–22 years. Initial data suggest that there is no increased risk from proton therapy, and initial dosimetric models propose a lower incidence of RIBT compared with photons; however, longer clinical follow-up is needed to quantify these risk reductions [22].

All of the risk reductions discussed in the preceding paragraphs is even more apparent with craniospinal irradiation (CSI) cases. Figure 1 shows a photon plan compared to a proton plan for a high-risk medulloblastoma craniospinal dose of 36 Gy. The proton plan is associated with virtually no dose to the normal tissues anterior to the spine.

## 3. Sarcoma

Proton therapy is a promising treatment option for sarcomas in the adolescent and young adult (AYA) population.

Sarcomas are rare but aggressive types of cancer that arise from bones, soft tissues, and cartilage. The incidence of sarcomas, relative to all cancers in the AYA population, is less than 10% in all subsets of the AYA age range of 15–39 years old. In the AYA population, there is a male:female preponderance of 2:1 [3].

Sarcomas that involve bone include chondrosarcomas (CS), Ewing’s sarcomas (ES), and osteosarcomas (OS). Sarcomas that include soft tissues, more commonly referred to as soft tissue sarcomas (STS) include rhabdomyosarcomas (RMS) and peripheral nerve sheath tumor (PSNT) sarcomas. There are also benign soft tissue sarcomas, such as desmoid tumors.

Unlike other tumors, which are managed almost exclusively by histology, sarcomas are also managed based on anatomic location. Base of skull chordomas (CH) and CS may be managed by CNS tumor specialists, including neurosurgeons and neuro-oncologists. Retroperitoneal sarcomas (RPS) may be managed by general surgical oncologists. ES and other extremity soft tissue sarcomas may be managed by orthopedic oncologists.

Base of skull (BOS) tumors are most commonly CH and CS. While the cornerstone of management of CH and CS is surgical resection, gross total resection (GTR) is difficult and postoperative RT is often recommended [23,24]. The dose–response gradient between 60 Gy and 70 Gy is quite steep, with control rates increasing from <25% to >80% [25,26,27,28,29,30,31,32]. The challenge of delivering 70 Gy to the BOS is made easier with PBT, as compared to XRT [33].

Para-spinal and retroperitoneal locations often require RT, for a variety of histologies, as this is a difficult area in which to obtain GTR with surgery. The use of PBT can avoid unnecessary radiation to organs located anteriorly, including the heart, lungs, liver, bowel and kidneys [34].

Within the pelvis, surgery can be morbid, and gross total resection can be difficult to obtain. Some studies have found similar outcomes between surgery and radiation, including a large study from the Children’s Oncology Group, which found no difference in local control or event-free survival between surgery and radiation, both 75% at 5 years. This also held true when accounting for tumor size less than or greater than 8 cm [35]. Other studies advocate for a dual-modality local control approach, consisting of both surgery and radiation, when managing large tumors [36]. Protons, as compared to advanced X-ray techniques, can spare critical pelvic structures. Examples of sparing have been demonstrated for ovaries, testes, bladder, and pelvic bones [37].

In summary, there is a strong case to be made for the use of proton RT in a wide variety of sarcomas. Anatomic location plays an important factor.

## 4. Breast Cancer

Breast cancer accounts for 30% of all cancers among AYA women, ranking it as the most common cancer type in AYA patients aged 15–39 years [38,39]. Moreover, the incidence of breast cancer among AYA patients has increased in the United States since 2004 [38,39]. AYA breast cancer patients are more likely to have unfavorable tumor biology, advanced disease, and poorer prognoses compared to older women [38]. For instance, approximately half of AYA breast cancer patients less than 30 years of age carry a germline mutation in *BRCA* or *TP53*, AYA breast cancer patients have a nine-fold higher local recurrence risk after breast conserving therapy compared to women over the age of 60, and 39% of AYA women with early-stage breast cancer are more likely to die compared to older women [40,41,42]. Additionally, AYA patients with breast cancer are more likely to endure psychosocial issues and experience more side effects from therapy [43]. As such, the multidisciplinary management of AYA patients with breast cancer also involves genetic counseling, fertility preservation, ovarian function management, nutrition, physical therapy and occupational therapy, psychosocial support, and survivorship assistance [38]. Given this information, radiation therapy strategies designed to decrease toxicity for AYA patients with breast cancer, such as PBT, are impactful in this population.

The dosimetric advantages of PBT can reduce the risk of cardiac toxicity, radiation pneumonitis, and radiation-associated secondary malignancies for AYA patients with breast cancer. The relationship between cardiac radiation dose and major coronary events is linear, with a 7.4% relative increase in major coronary events for each 1-Gy increase in mean heart dose [44]. This translates to an estimated 0.3% to 0.6% per 1-Gy absolute increase in major coronary events by the age of 80 depending on patient age at time of treatment and cardiac risk factors [44]. Furthermore, there is no lower threshold where there is an absence of risk [44]. Other studies have also reported radiation dose to the left ventricle and coronary artery subsegments correlate with major coronary events [45,46,47]. As such, radiation techniques designed to minimize cardiac and coronary artery dose as much as possible are needed, especially in patients with baseline cardiac risk factors, young patients, and patients who have received cardiotoxic systemic agents. Several studies have demonstrated that PBT reduces radiation dose to the heart compared to intensity-modulated radiation therapy (IMRT) and three-dimensional conformal radiation therapy (3DCRT) [48,49,50,51,52].

Dosimetric risk factors for radiation pneumonitis are ipsilateral lung V20Gy >30% and ipsilateral mean lung dose >15 Gy [53]. PBT reduces radiation dose to the lungs in patients with early-stage and locally advanced breast cancer compared to IMRT and 3DCRT, translating to a reduced risk of radiation pneumonitis [54,55,56]. Patients with underlying pulmonary comorbidities that increase the risk of radiation pneumonitis, such as interstitial lung disease, and patients receiving systemic therapy agents associated with pneumonitis may benefit more from reduced radiation lung dose [57,58] Radiation-associated secondary malignancies in this population include angiosarcoma, contralateral breast cancer and lung cancer. Reduced risk of secondary malignancy is especially important in AYA breast cancer patients who harbor germline mutations [59] Grantzau et al. reported the risk of secondary lung cancer at least 5 years after radiation therapy for breast cancer increases linearly by 8.5% per Gray [60]. The risk is higher at 17.5% per Gray in those with smoking histories [60]. Stovall et al. reported women less than 40 years of age with contralateral breast quadrants exposed to >1.0 mean Gy had a 2.5 times higher risk of contralateral breast cancer compared to unexposed women [61]. As such, radiation approaches designed to deliver less dose to adjacent normal tissues, such as PBT, are essential in minimizing the risk of secondary cancers for AYA breast cancer patients. Modeling studies suggest a lower risk of radiation-associated secondary cancers after PBT compared with modern XRT techniques [62,63].

Given the potential for PBT to reduce radiation dose to normal tissues, thereby potentially reducing the risk of radiation-associated toxicities, the indications for PBT in AYA breast cancer patients are vast. Clinical evidence and indications for PBT with breast cancer include regional nodal radiation (Figure 2), whole breast radiation, partial breast radiation, bilateral breast radiation, patients with and without reconstruction, and reirradiation cases as outlined in the Consensus Statement from the particle Therapy Cooperative Group Breast Cancer Subcommittee [64]. The primary limiting factor with PBT studies so far for breast cancer is the absence of long-term clinical data clearly demonstrating improvement in the therapeutic ratio for PBT. Large prospective studies are needed for reliable comparisons of radiation-associated toxicities across treatment modalities. These data will continue to become available in the years ahead from ongoing randomized trials and institutional reports evaluating the outcomes of PBT.

## 5. Hodgkin Lymphoma

Contrary to most adult and pediatric cancers, Hodgkin Lymphoma (HL) predominately occurs in the AYA population. It is the most frequently diagnosed cancer in patients aged 15–19 years [65]. Survival for AYA HL patients is excellent, with 10-year rates exceeding 90% [66]. Given the excellent survival outcomes, the treatment paradigm for HL now focuses on minimizing treatment intensity and late effects. HL survivors experience premature mortality due to treatment-associated morbidities, including cardiac disease, pulmonary disease, and secondary cancers.

It is well established that radiation field extent and radiation dose delivered correlate with treatment-associated morbidities for HL survivors [67]. In a cohort of 2985 5-year HL survivors <21 years at diagnosis and treated from 1970 to 1999, cardiac toxicities decreased over the study period corresponding to the overall decline in cardiac radiation therapy exposure [68]. The proportion of patients treated with mean heart doses ≥35 Gy decreased from 60% in the 1970s to 3% in the 1990s and the percentage of patients who received no cardiac radiation increased from 6% in the 1970s to 52% in the 1990s [68]. This correlates with the evolution of HL radiation treatment approaches, which primarily consisted of >36 Gy mantle field radiation prior to 1990 versus 20–36 Gy involved field radiation therapy after 1990 [67]. Concurrently, the incidence of coronary artery disease declined significantly with treatment era (hazard ratio: 0.44 for 1990s versus 1970s) after factoring in cardiotoxic chemotherapy suggesting the reduction in mean heart dose contributed significantly to the decline in coronary artery disease for HL survivors [68]. This confirms that every effort to reduce radiation dose to the heart and cardiac substructures is essential for AYA HL patients treated with radiation therapy.

Female HL survivors carry an elevated risk of developing a secondary breast cancer. Early HL survivorship studies reported 30-year cumulative secondary breast cancer incidences as high as 30% [69,70,71]. This is likely a consequence of the large volume of breast tissue exposed to high radiation doses with historic mantle fields. A cohort study of 3905 5-year HL survivors aged 15–50 years at time of treatment from 1965 to 2000 from the Netherlands reported a lower risk of secondary breast cancer after radiation therapy that excluded axillae compared to mantle radiation therapy suggesting that a reduction in radiation exposure of both breast and axillary tissues is essential in minimizing secondary breast cancer risk [72]. More recently, the British Columbia Cancer Agency Lymphoid Cancer Database investigated the incidence of secondary breast cancer in a cohort of 734 female HL survivors ≤50 years of age at time of treatment from 1961 to 2009 [73]. Patients were grouped according to radiation therapy field side as mantle radiation (*n* = 231), small field radiation therapy (defined as involved field, involved site, or involved nodal; *n* = 185), or chemotherapy only (*n* = 318) [73]. The median radiation therapy dose was 35 Gy (range: 15.8–52 Gy) [73]. With a median follow-up of 18 years, 54 patients (7%) developed secondary breast cancer [73]. The 20-year cumulative incidence of secondary breast cancer was 7.5% after mantle radiation therapy, 3.1% after small field radiation therapy, and 2.0% after chemotherapy only [73]. This confirms that the risk of secondary breast cancer is significantly elevated after mantle radiation therapy but suggests that the risk with modern small field radiation therapy techniques is not greater than after chemotherapy alone. Taken together, the data demonstrate that any effort to reduce radiation dose to breast and axillary tissues in female HL patients is clinically meaningful.

The International Lymphoma Radiation Oncology Group Consensus Guidelines acknowledge patients with mediastinal lymphomas likely derive a great benefit from proton therapy. Patients and scenarios most likely to derive the greatest benefit include: (1) patients with mediastinal disease that extends below the origin of the left main stem coronary artery and is anterior to, posterior to, or on the left side of the heart; (2) young female patients to reduce radiation dose to the breast tissue; and (3) heavily pretreated patients who are at high risk of radiation-associated toxicities to the bone marrow, heart, and lungs [74]. Figure 3 illustrates a proton radiation therapy plan for a 16-year-old female with bulky mediastinal HL treated to 21 Gy. The mean breast doses were 1.6 and 0.41 Gy. The mean heart dose was 2.6 Gy. For a similar case of mediastinum and left axilla radiation presented in the International Lymphoma Radiation Oncology Group Consensus Guidelines, the mean reported heart dose with IMRT was 14.6 versus 9.9 Gy with protons (32% reduction) [74]. The mean breast dose with IMRT was 6.7 versus 2.6 Gy with protons (61% reduction) [74].

The philosophy of treatment de-intensification for HL will only continue given the excellent survival rates in this population. Current toxicity data related to radiation therapy approaches reflect treatment paradigms from the era of large treatment fields, high radiation doses, and use of XRT. Nevertheless, these data illustrate every effort to reduce radiation dose to normal tissues, and the volume of normal tissue exposed to radiation is essential in minimizing radiation-associated toxicities, and PBT will likely be crucial in accomplishing this. As more HL patients are treated with proton therapy, data regarding PBT from the modern era will emerge and help inform future trials and contribute to the overarching goal of maximizing cure and minimizing toxicities for HL patients.

## 6. Miscellaneous Cancers

Head and neck malignancies are very rare in the AYA population but can include soft tissue, bone, human papilloma virus (HPV)-positive orpharyngeal, and nasopharyngeal tumors. Radiation therapy is associated with cataract formation, hearing loss, trismus, neck fibrosis and xerostomia [75]. These sequalae negatively impact quality of life for AYA survivors. Hamilton et al. surveyed 36 AYA head and neck cancer survivors to assess treatment-related sequalae. Fifty-one percent of participants report severe symptoms including xerostomia, dental/mucosal sensitivity, and dysphagia. Additionally, 35% had moderate/high anxiety scores and 48% had moderate/high depression scores [76]. These results indicate that AYA head and neck cancer survivors have seginitivate late effects from radiation therapy that negatively impact their quality of life. As such, PBT may help reduce the risk of radiation-associated sequalae and consequently improve patient-reported outcomes for AYA patients. Numerous studies describing PBT for head and neck cancers are emerging. Overall, there is no difference in oncologic outcomes between PBT and IMRT. PBT has been associated with reduced symptom burden during the subacute recovery phase after treatment [77]. For example, a case-matched analysis of 50 patients treated with PBT for oropharyngeal carcinoma and matched with 100 controls treated with IMRT demonstrated that patients treated with IMRT had 20% more grade 3 weight loss (compared to baseline) and more gastrostomy tubes 3 months and 1 year after treatment [78]. Romesser et al. demonstrated lower rates of grade 2 or higher dysgeusia with PBT compared to IMRT in a cohort of 41 patients: 5.6% versus 65.2%. PBT was also associated with lower rates of mucositis (16.7% versus 52.2%) and nausea (11.1% versus 56.5%) compared to IMRT [79]. Further research efforts are needed to evaluate differences in long-term radiation-associated adverse events for patients with head and neck malignancies [80].

Testicular cancer is the most common cancer diagnosed in men less that 40 years of age [81]. All patients are treated with an orchiectomy and then receive adjuvant therapy consisting of chemotherapy and/or radiation therapy depending on risk factors. Radiation therapy, if used, treats the lymph nodes involving the pelvis and para-aortic regions. Patients with testicular cancer have excellent long-term survival rates, which predisposes them to radiation therapy-related morbidity related to normal tissue function and risk of secondary malignancies. In a study of 2629 seminoma patients from 12 cancer centers, 468 second cancers were identified, corresponding to a standardized incidence ratio of 1.61 [82]. PBT can theoretically reduce the dose associated to the heart, lungs, and abdomino-pelvic spaces. Pasalic et al. conducted an observational study of 55 patients with testicular seminoma treated with radiation therapy. Forty-four patients received photon treatment and 11 patients received PBT. PBT was associated with lower rates of diarrhea compared to photon therapy: 0% versus 29.5%. With a median follow-up of 61 months, three patients in the photon arm were diagnosed with a secondary malignancy. There were no secondary malignancies in the group of patients treated with PBT [83]. Other reports have also suggested a lower secondary malignancy rate for seminoma patients treated with PBT compared to photon therapy [84].

## 7. Conclusions

This report highlights the potential utility of PBT for certain AYA cancers, especially with reducing radiation doses to organs at risk and thereby potentially lowering risks of certain treatment-associated toxicities. The dosimetric advantages of reducing radiation dose to normal tissues with PBT can be clinically significant and highly desirable to optimize the therapeutic ratio for these patients given the high cure rates as well as high rates of survivorship challenges. Additional information and understanding of which cases are most likely to derive the greatest benefit from PBT and updated outcomes of PBT for AYA cancers are still needed. Development of and access to clinical trials for AYA patients, especially with regard to proton therapy, will be beneficial. Additionally, establishment of working groups within national and international societies to help advocate for and advance AYA care, including quality of life outcomes, will also be useful.

## Figures and Tables

**Figure 1 cancers-15-04269-f001:**
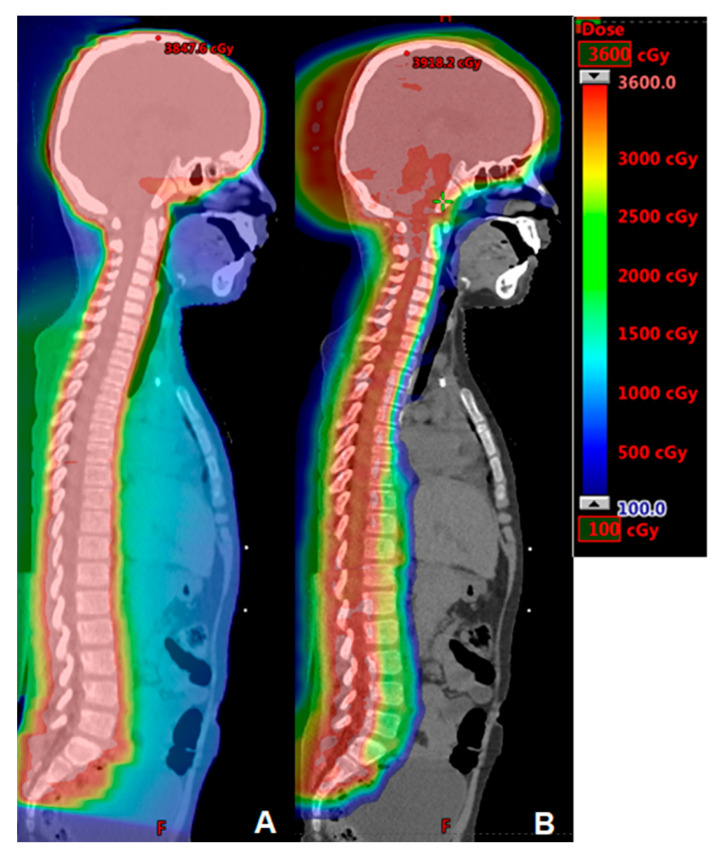
Comparison IMRT (**A**) and proton (**B**) plans for craniospinal irradiation in a high-risk medulloblastoma case treated to 36 Gy.

**Figure 2 cancers-15-04269-f002:**
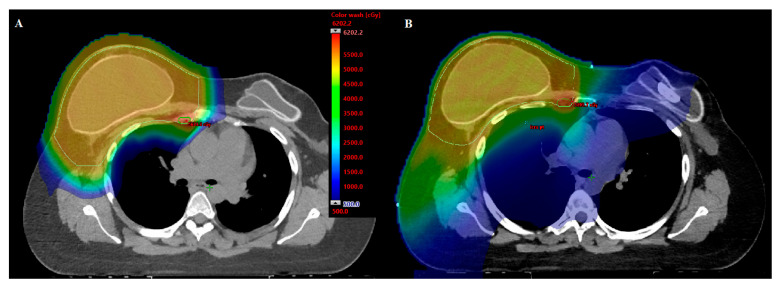
Comparison proton (**A**) and IMRT (**B**) plans for right chestwall, regional nodal, and internal mammary nodal boost irradiation plans for 28-year-old woman with cT3N3M0 triple negative, grade 3, invasive ductal carcinoma. Mean heart dose with IMRT is 6.99 Gy versus 1.0 Gy with protons (86% reduction in dose).

**Figure 3 cancers-15-04269-f003:**
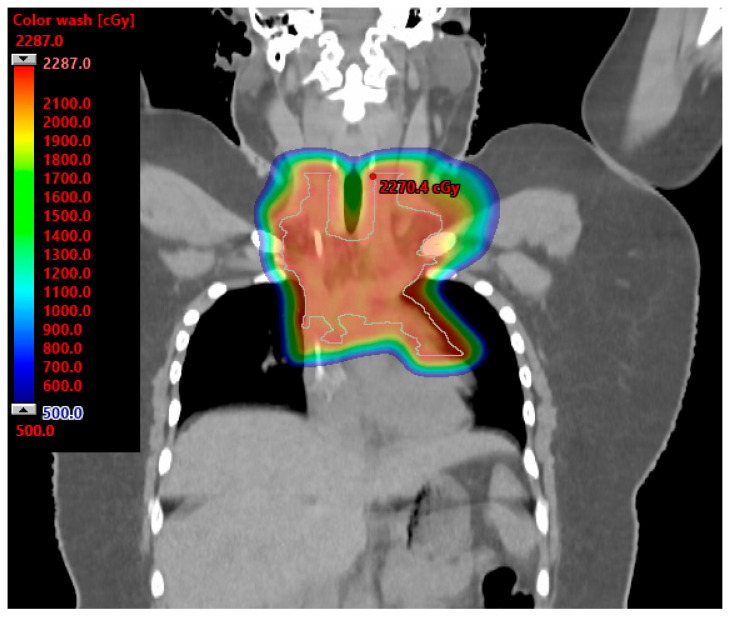
Proton plan for 16-year-old female with large mediastinal adenopathy treated to 21 Gy. The mean breast doses were 1.6 and 0.41 Gy. The mean heart dose was 2.6 Gy.

## Data Availability

The data presented in this study are available in this article.

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
