# Peer review of "Proton Therapy in the Adolescent and Young Adult Population"

_cancers, 2023, doi:10.3390/cancers15174269_

Round 1

Author Response

In children is clear that proton therapy improves the dose distribution to normal tissues in comparison to photon treatment. Frappaz et al found that up to 40% of the patients discussed within the Adolescent and Young Adults (AYAS) brain tumor national Web conference were presenting medulloblastoma (Bulletin du Cancer Volume 103, Issue 12, December 2016, Pages 1050-1056). I would put a picture showing a CSI proton irradiation planning vs photon planning

This picture has been added as Figure 1.

  • In the clinical indications, I would discuss head and neck cancers too. Despite the survival rate is excellent in AYAs, however patients undergoing H&N radiotherapy are at high risk of developing significant long-term health effects from treatment, such as xerostomia, dysgeusia, dysphagia, and neck fibrosis, which can have a considerable impact on quality of life.

This has been added under “Miscellaneous Tumors.”

  • Conclusions: I would really stress the following points
  • 1) In the future we have to increase the portfolio of trials for AYAS as well as the accessibility to proton therapy management
  • 2) We need a working group within the proton therapy community, as already running in the several pediatric oncology societies, to evaluate outcome but also quality of life

These points have been added to the end of the conclusions.

Reviewer 2 Report

Title: Proton Therapy in the Adolescent and Young Adult Population

 Reviewer comments

General:

The authors give an overview over the potentials of proton therapy in the younger non pediatric population.

The overview is well written and gives a comprehensive overview for the general reader.

It would be nice to add a paragraph on germ cell tumors like seminoma testis.

 I have only minor suggestions for consideration.

 Page 2 line 62

“PBT appears to reduce these risks” it would be nice to add a number quantifying this.

Page 2 lines 64-65

“Patients treated with PBT also had a greater height standard deviation score.” It is not exactly clear what the authors mean with this sentence. Are PBT patients higher/longer than XRT patients?

Page 2 lines 65-66

“There was no significant difference in the incidence of precocious puberty, adrenal insufficiency or growth hormone deficiency.”

I assume this is between PBT and XRT, but it misses in the sentence.

Page 2 lines 67-69

“PBT appears to lower the risk of primary hypothyroidism, compared to XRT CSI. This is felt to be due to lower dose exposure to the thyroid gland and the pituitary gland.” For the pituitary gland this is probably true for the boost part and not so much for the CSI.

Page 2 lines 70-73

“With XRT, the relative risk of RIBT is between 6 and 10 with a latency period of 6-22 years” In a paper about advantages of protons its not sufficient to just state the values for photons.

Page 3 lines 125-127

“Given this information, radiation therapy strategies designed to increase the therapeutic ratio for AYA patients with breast cancer, such as PBT, are impactful in this population.”

Therapeutic radio is either more cure or less toxicity. In breast cancer it is not assumed that dose is the issue. For the general reader perhaps more clear to specify that the aim is less toxicity.

Author Response

It would be nice to add a paragraph on germ cell tumors like seminoma testis.

This has been added under “Miscellaneous Tumors.”

  Page 2 line 62

 “PBT appears to reduce these risks” it would be nice to add a number quantifying this.

The numbers have been added: “PBT appears to reduce these risks. Eaton et al published that PBT decreased risks of hypothyroidism (23% versus 69% with photon therapy), sex hormone deficiency (3% versus 19% with photon therapy), and requirement for any endocrine replacement therapy (55% versus 78% with photon therapy).”

Page 2 lines 64-65

“Patients treated with PBT also had a greater height standard deviation score.” It is not exactly clear what the authors mean with this sentence. Are PBT patients higher/longer than XRT patients?

That is correct. This has been updated.

Page 2 lines 65-66

There was no significant difference in the incidence of precocious puberty, adrenal insufficiency or growth hormone deficiency.”

I assume this is between PBT and XRT, but it misses in the sentence.

That is correct. This has been updated.

Page 2 lines 67-69

“PBT appears to lower the risk of primary hypothyroidism, compared to XRT CSI. This is felt to be due to lower dose exposure to the thyroid gland and the pituitary gland.” For the pituitary gland this is probably true for the boost part and not so much for the CSI.

That is correct. This has been updated to state after spine and boost radiation as per the authors conclusions in the paper.

Page 2 lines 70-73

“With XRT, the relative risk of RIBT is between 6 and 10 with a latency period of 6-22 years” In a paper about advantages of protons its not sufficient to just state the values for photons.

This is an excellent point, however, the literature does not provide exact numbers for protons yet as we need longer term follow up to determine this. The sentence has been updated as: Initial data suggests there is no increased risk from proton therapy and initial dosimetric models propose a lower incidence of RIBT compared with photons, however, longer clinical follow up is needed to quantify these risk reductions.

Page 3 lines 125-127

“Given this information, radiation therapy strategies designed to increase the therapeutic ratio for AYA patients with breast cancer, such as PBT, are impactful in this population.”

Therapeutic radio is either more cure or less toxicity. In breast cancer it is not assumed that dose is the issue. For the general reader perhaps more clear to specify that the aim is less toxicity.

The sentence has been updated to state “decrease toxicity.”

Round 2

Reviewer 1 Report

no comments